# Mitochondrial Reactive Oxygen Species in Infection and Immunity

**DOI:** 10.3390/biom14060670

**Published:** 2024-06-08

**Authors:** Arunima Mukherjee, Krishna Kanta Ghosh, Sabyasachi Chakrabortty, Balázs Gulyás, Parasuraman Padmanabhan, Writoban Basu Ball

**Affiliations:** 1Department of Biological Sciences, School of Engineering and Sciences, SRM University AP Andhra Pradesh, Guntur 522502, Andhra Pradesh, India; arunima_m@srmap.edu.in; 2Lee Kong Chian School of Medicine, Nanyang Technological University Singapore, 59 Nanyang Drive, Singapore 636921, Singapore; gkkanta@ntu.edu.sg (K.K.G.); balazs.gulyas@ntu.edu.sg (B.G.); 3Department of Chemistry, School of Engineering and Sciences, SRM University AP Andhra Pradesh, Guntur 522502, Andhra Pradesh, India; sabyasachi.c@srmap.edu.in; 4Cognitive Neuroimaging Centre, 59 Nanyang Drive, Nanyang Technological University, Singapore 636921, Singapore; 5Department of Clinical Neuroscience, Karolinska Institute, 17176 Stockholm, Sweden

**Keywords:** mitochondrial reactive oxygen species, electron transport chain, inflammasome, bacteria, protozoa, virus, fungi

## Abstract

Reactive oxygen species (ROS) contain at least one oxygen atom and one or more unpaired electrons and include singlet oxygen, superoxide anion radical, hydroxyl radical, hydroperoxyl radical, and free nitrogen radicals. Intracellular ROS can be formed as a consequence of several factors, including ultra-violet (UV) radiation, electron leakage during aerobic respiration, inflammatory responses mediated by macrophages, and other external stimuli or stress. The enhanced production of ROS is termed oxidative stress and this leads to cellular damage, such as protein carbonylation, lipid peroxidation, deoxyribonucleic acid (DNA) damage, and base modifications. This damage may manifest in various pathological states, including ageing, cancer, neurological diseases, and metabolic disorders like diabetes. On the other hand, the optimum levels of ROS have been implicated in the regulation of many important physiological processes. For example, the ROS generated in the mitochondria (mitochondrial ROS or mt-ROS), as a byproduct of the electron transport chain (ETC), participate in a plethora of physiological functions, which include ageing, cell growth, cell proliferation, and immune response and regulation. In this current review, we will focus on the mechanisms by which mt-ROS regulate different pathways of host immune responses in the context of infection by bacteria, protozoan parasites, viruses, and fungi. We will also discuss how these pathogens, in turn, modulate mt-ROS to evade host immunity. We will conclude by briefly giving an overview of the potential therapeutic approaches involving mt-ROS in infectious diseases.

## 1. Introduction

Reactive oxygen species (ROS) are formed due to the transfer of electrons to molecular oxygen from compounds which are highly reducing in nature. This transfer of electrons results in two forms of ROS—a highly reactive free radical ROS, like the hydroxyl radical (HO˙), and a less reactive nonradical ROS, such as hydrogen peroxide (H_2_O_2_) [1]. ROS is usually associated with cellular damage and toxicity, which includes both oxygen free radicals and nonradical oxidants. However, regulated levels of ROS are crucial for a wide range of physiological processes where the oxidative modification of biological macromolecules manifests in variety of cellular functions, including metabolism, transcriptional regulation, cellular replication, phosphorylation/dephosphorylation, development, apoptosis, ageing, steroidogenesis, temperature regulation, proliferation, and immunity [2,3,4,5].

In general, mitochondria are considered to be the major sites for ROS generation, although a significant amount of ROS may be generated in the cytoplasm, cell membrane, or different cellular compartments, including peroxisomes and endoplasmic reticulum (ER), when induced by different stimulants such as pathogens, nanoparticles, radiation, or photoactivation [3,5,6,7,8,9,10,11]. Cytoplasmic ROS (cyto-ROS) are generated majorly by the enzymes belonging to the family NADPH oxidase (NOX), whereby they transfer electrons from NADPH to molecular oxygen to generate superoxide and, subsequently, other ROS molecules (Figure 1). In mammals, in addition to NOX, different isoforms of nitric oxide synthase (NOS), like neuronal NOS (nNOS/NOSI), inducible NOS (iNOS/NOSII), and endothelial NOS (eNOS/NOSII), also contribute to the production of cyto-ROS [8,12]. The first characterized NOX was NOX-2, which consists of six subunits and two integral membrane proteins/subunits, the α subunit (p22^phox^) and β subunit (gp91^phox^). Along with these subunits there are regulatory subunits, namely p67^phox^, p40^phox^, and p47^phox^, existing as a cytosolic complex. In neutrophils, p47^phox^ is phosphorylated upon stimulation and enhances the binding of p67^phox^ and Rac protein, which facilitates the translocation of the complex to the membrane and the activation of oxidase [13,14]. Hydrogen peroxide (H_2_O_2_) is also produced in the endoplasmic reticulum (ER) as a byproduct of oxidative protein folding and disulfide bond formation in folded proteins (Figure 1). Additionally, the dysregulation of disulfide bonds in the stressed ER leads to the accumulation of ROS, causing oxidative stress. ER-ROS is generated by ER-confined NOX4 and the monooxygenase system is associated with the membrane [8,15]. While mitochondrial ROS (mt-ROS) is chiefly formed as a byproduct of oxidative phosphorylation (OXPHOS), in peroxisomes, electron transfer does not yield adenosine triphosphate (ATP )but generates H_2_O_2_, where a variety of oxidases catalyze the peroxisomal ROS generation [8,16]. Under various physiological or pathological conditions, these organelles contribute to the production of ROS to a varying degree, each with their own significance in cellular physiology [17,18,19]. Additionally, red blood cells (RBCs), in spite of the absence of mitochondria, can produce ROS through heme-bound iron (Fe^2+^), which, when it interacts with oxygen (O_2_), yields superoxide (Figure 1) [20].

However, elevated levels of ROS are associated with a wide range of diseases, like cancer, neurodegeneration, and cardiac disorders [21,22,23]. Likewise, an excess of mitochondrial ROS (mt-ROS) is also implicated in a plethora of diseases and conditions, such as ischemia and tumorigenesis [24,25]. The importance of countering oxidative stress is evidenced by the antioxidant defense machinery employed by the cells. The antioxidant defense molecules can be categorized into three distinct groups—the first comprises the non-enzymatic metabolites like glutathione, tocopherol, and ascorbate, to name a few. The second group comprises superoxide dismutases (SODs), which include Cu-SOD, Zn-SOD, Mn-SOD, and Ni-SOD. The last group consists of peroxidase enzymes like ascorbate peroxidase, cytochrome c peroxidase, glutathione peroxidase, and catalase. Although the mitigation of excess ROS levels is physiologically important, it has been shown that regulated elevation of mt-ROS helps in antiviral and antibacterial immunity. mt-ROS also help in the maintenance of overall cellular homeostasis by functioning as a signaling cue for apoptosis, cell proliferation and differentiation, and insulin secretion [26,27,28,29,30]. In this review, we are only going to focus on the different roles of mt-ROS in the context of pathogenic infections.

## 2. Generation and Regulation of Mitochondrial ROS

Mitochondria, a double membrane-bound, energy-harvesting organelle, plays critical roles in carrying out metabolic cycles, maintaining ion balance, inducing autophagy, and eliciting immune response against infections. One of the major ways by which mitochondria activates immune signaling involves the generation of ROS [31].

An obvious pathway of ROS production in mitochondria involves the leakage of electrons from the electron transport chain (ETC) complexes and their subsequent interaction with molecular oxygen to form the superoxide anion, which is then catalytically converted to H_2_O_2_ by mitochondrial-specific superoxide dismutase (SOD) (Figure 2) [32,33]. Flavoproteins, ubiquinone, iron–sulfur proteins, and cytochromes constitute the electron carrier centers embedded in the four ETC protein complexes [32]. The four ETC complexes, along with complex V, also known as ATP synthase, further assemble in different stoichiometries to form mitochondrial respiratory chain (MRC) supercomplexes to complete OXPHOS and generate ATP. Any disruption in this higher order supramolecular assembly of supercomplexes causes increased levels of mt-ROS production, as evident from a previous study conducted in neurons [34]. In the ETC, reduced NADH is oxidized by complex I (NADH ubiquinone oxidoreductase), and, similarly, Complex II or succinate dehydrogenase oxidizes succinate in the Krebs cycle, forming fumarate and reduced FADH_2_. The electrons from complex I and complex II move through the electron carriers to ubiquinone (Q) to form ubiquinol (QH_2_), which, in turn, is oxidized by complex III or cytochrome c reductase. Subsequently, the electrons from reduced complex III are passed to complex IV or cytochrome c oxidase via the electron carrier cytochrome c. Complex IV transfers electrons to the molecular oxygen, the terminal electron acceptor of the aerobic respiration, producing water (Figure 2). This electron flow follows an energetically favorable transfer of electrons from electron donors to electron acceptors and is coupled with proton pumping to the inter membrane space (IMS), creating a proton gradient across the inner mitochondrial membrane (IMM). This electrochemical gradient is coupled to complex V, whereby the free energy change by the flow of protons to the matrix from IMS through complex V produces ATP. In the Q pool, the electrons may also be contributed by other enzymes of the IMM, including sn-glycerol-3-phosphate dehydrogenase (G3PDH), sulfide:quinone oxidoreductase (SQR), proline dehydrogenase, electron transfer flavoprotein oxidoreductase (ETFQO), and dihydroorotate dehydrogenase [6,35].

### 2.1. Sites and Regulation of mt-ROS Generation

It has been estimated that about 0.2–2% of the electrons from the ETC contribute to the production of mt-ROS [36]. However, this is the estimate for the resting state of mitochondria, when ATP demand is low; the ROS production is negligible in active mitochondria, where more ATP is generated. This eliminates the misconception that higher mitochondrial respiration will lead to higher ROS generation, as, in active respiration, ETC components work more efficiently and are in an oxidized state, which diminishes ROS production. On the contrary, when the ATP demand is reduced, ETC is slowed down and an accumulation of NADH or ubiquinol occurs. This reduced NAD or quinone pool, consequently, leads to ROS generation. Hence, it can be stated that ROS production is dependent on the redox balance of mitochondrial substrates and the redox pools of NAD and quinone, which aid in ROS generation at different sites [6]. Eleven sites of ROS production pertaining to ETC, OXPHOS, and substrate catabolism have been reported; of these, six sites are associated with the NADH/NAD^+^ isopotential pool, while the other five sites are associated with the ubiquinol/ubiquinone (QH_2_/Q) isopotential pool. Superoxides are generated by the addition of a single electron to molecular oxygen, whereas H_2_O_2_ generation requires two electrons. These sites have been dealt with in greater detail in a review article by Martin Brand, which the readers of this article are encouraged to go through [37]. Complex I comprises two arms—a matrix arm and an IMM arm. The membrane arm consists of co-factors like flavin mononucleotides (FMN) and various iron–sulfur clusters. The flavin-containing site (I_F_) and the quinone-binding site (I_Q_) of complex I are equally capable of generating ROS in the matrix during the electron transfer from NADH to Q. Inhibitors of the I_Q_ site block the electron transfer to Q, and ROS production is increased at the I_F_ site. Initially, complex II was not considered to be a significant contributor of ROS because, under physiological conditions, the ROS generated by complex II was negligible. However, it was reported that, in rat skeletal muscle mitochondria, in the absence of complex I and III, and at a lower concentration of succinate, complex II could generate high rates of superoxide and H_2_O_2_. Complex II can generate ROS bidirectionally, for instance, by utilizing the electrons from succinate in the forward reaction and, in the reverse direction, by utilizing the electrons from the reduced ubiquinol pool. Complex II consists of four subunits; SDHA, the flavoprotein subunit, containing covalently bound flavin adenine dinucleotide (FAD) in the active site, which aids in the removal of electrons from succinate; SDHB, the iron–sulfur protein subunit, containing a chain of three iron–sulfur clusters, [2Fe-2S], [4Fe-4S], and [3Fe-4S]; and two transmembrane cytochrome b heme subunits, SDHC and SDHD. When succinate is oxidized, two electrons move to the flavin at site II_F_, then pass one at a time through the Fe-S clusters to reduce ubiquinone to ubiquinol at site II_Q_. Flavoprotein is found in the IMM on the matrix side and FAD, being a potent site for the loss of electrons, results in the generation of ROS. Complex III-derived mt-ROS plays a crucial role in redox signaling. Complex III comprises cytochrome b, cytochrome c1, and a 2Fe-2S cluster wrapped by an iron–sulfur protein, which has high potential. Cytochrome b in the IMM consists of two ubiquinone-binding sites with different potentials. The first one is the QH_2_ oxidation site (Q_o_) and the other one is the Q reduction site (Q_i_). In complex III, electrons are passed through the Q cycle. During this process, at the Q_o_ site, ubisemiquinone (QH^−^), which carries a single electron, is capable of moving freely in complex III and can leak the electron directly to oxygen-producing ROS in a nonenzymatic manner. Although Q_o_ is located in the IMS side of the IMM, the generated ROS can seep into both the IMS and the matrix. Two different explanations have been put forward—(i) an equal proclivity of the semiquinone to disseminate on both side of the membrane, and (ii) the utilization of the hydrophobic part of the Q_o_ site for the reaction of neutral semiquinone with oxygen. When the Q_i_ site is blocked, the electrons halt at the III_Qo_ site and, consequentially, it reacts with O_2_ and generates ROS (Figure 2) [36,38,39,40,41].

The NADH/NAD^+^ pool may acquire electrons from different substrates, such as the 2-oxoglutarate dehydrogenase complex (OGDHC), the pyruvate dehydrogenase complex (PDHC), the branched-chain 2-oxoacid dehydrogenase complex (BCOADHC), or the 2-oxoadipate dehydrogenase complex (OADHC). Similarly, the QH_2_/Q pool also acquires electrons from mitochondrial glycerol-3-phosphate dehydrogenase (mGPDH), the electron-transferring flavoprotein (ETF):Q oxidoreductase system (ETF:QOR), or dihydroorotate dehydrogenase (DHODH), apart from succinate. The substrate that is oxidized decides the contribution of specific sites in ROS generation [37,42]. Another source of mt-ROS generation is monoamine oxidase (MAO). MAO exists in two isoforms, namely MAO-A and MAO-B, which are responsible for the metabolism of various neurotransmitters such as serotonin, norepinephrine, benzylamine, phenylethylamine, and dopamine. MAO generates H_2_O_2_ as a byproduct of oxidative deamination in the outer mitochondrial membrane (OMM). H_2_O_2_ generated by oxidative deamination is significantly higher than that from the oxidation of succinate by complex II [43]. Additionally, cytoplasmic enzyme NOX-2 also contributes to mt-ROS generation by inducing ROS generation in the mitochondria, in a process called ROS-induced ROS (RIR) [5].

### 2.2. Regulation of mt-ROS

Other than the above-mentioned conditions and requirements, ROS generation depends upon the concentration of oxygen, the availability and concentration of the electron carrier capable of transferring electrons to oxygen, and the concentration of carriers which can undergo autoxidation and release an electron to form a radical species. However, when the mitochondrial carriers are considered, they differ on the basis of species, tissue, hormonal balance, and age. Also, the type of substrate utilized should also be taken into consideration. For instance, pyruvate dehydrogenase (PDH) and α-ketoglutarate dehydrogenase (KGDH) can produce four times and eight times more ROS, respectively, than complex I in skeletal muscle mitochondria [44,45].

### 2.3. Regulators and Inhibitors of mt-ROS

In biological systems, superoxide anion and H_2_O_2_ are considered to be of great importance, as the majority of ROS are derived from these two primary and significant mitochondrial ROS. The superoxide, which is formed as a byproduct of cellular respiration, may give rise to hydroperoxyl radical by reacting with water, or it may further react with a nitric oxide radical to give rise to highly toxic reactive nitrogen species (RNS).To keep a check on the levels of ROS production, there exists an antioxidant defense system. Generally, as mentioned earlier, SOD catalyzes the conversion of superoxide to H_2_O_2_, which is stable to a greater extent than superoxide and is critical for redox signaling. Two types of SOD enzymes are found in mitochondria. Manganese-SOD (Mn-SOD) is present in the matrix of mitochondria, while the copper/zinc SOD (Cu/Zn-SOD) is present in the IMS. Although H_2_O_2_ is a critical signaling molecule, an excess of H_2_O_2_ also leads to cellular damage. To scavenge the excess H_2_O_2_, peroxiredoxin (PRx), glutathione peroxidase (GPx) (in association with other antioxidant molecules, namely glutathione (GSH, reduced form) and thioredoxin), and catalase systems are present in the mitochondria. GPx can neutralize a wide variety of toxins, including the one formed in lipid peroxidation, by utilizing GSH as an electron donor and forming glutathione disulfide (GSSG, oxidized form). Subsequently, flavoenzyme GSH reductase and NADPH regenerate GSH from GSSG to maintain the balance of reduced vs. oxidized glutathione for the proper functioning of this antioxidant defense system. GPx has six isoenzymes in mammalian cells, of which GPx-1 is expressed nonuniformly in all tissues. It is chiefly localized in cytosol; however, the mitochondrial matrix contains about 30% GPx-1, which provides crucial protection against oxidative damage against the ROS produced in the mitochondria. The cells with elevated levels of GPx-1 indeed had reduced ROS levels but showed lower mitochondrial membrane potential. This, therefore, establishes the fact that GPx regulates mt-ROS levels by concomitantly modulating the mitochondrial potential and mitochondrial uncoupling. Also, it could be inferred that higher GPx levels would lead to a reducing environment. These findings were supported by the experiments carried out in guinea pig’s heart mitochondria, where the use of uncouplers led to a lower membrane potential and, subsequently, lower ROS production. However, it has also been found that higher uncoupling may lead to ROS accumulation. Considering these contradictory observations, Aon et al. have given a redox optimized ROS balance hypothesis, according to which, at the maximal redox potential of either the ETC or the ROS scavenging couples, the homeostasis of ROS is interrupted, which elevates its production. For instance, in a higher oxidizing environment, the antioxidants do not function properly and ROS production may be enhanced. On the contrary, at a highly reduced potential, ROS production may bypass scavenging. Hence, the redox potential of mitochondria should be maintained at an optimum level to avoid oxidative stress. At extreme levels of potential, pathological conditions may be aided, which is in agreement with the obtained result to an extent, as the overexpression of GPx-1 also led to insulin resistance, obesity, and the inhibition of insulin-mediated activation of the Akt pathway in mice.

Catalase is another enzyme which efficiently degrades H_2_O_2_. Although predominantly localized in peroxisomes, the presence of mitochondrial catalase in rat hearts has been reported. When catalase was overexpressed in the mitochondrial compartment of HepG2 cells, it was capable of protecting the cells from the cytotoxicity caused by the elevated levels of H_2_O_2_ induced by antimycin-A, an inhibitor of complex III. PRx belongs to the peroxidase family of enzymes, which are highly conserved. Different organisms have different numbers of isoforms of the PRx enzyme. In *Saccharomyces cerevisiae*, PRxs have five isoforms, of which 1-Cys mTpx is present in mitochondria. Meanwhile, in mammalian cells, *E. coli*, and *Arabidopsis*, six, three, and nine isoforms of PRx enzymes have been reported, respectively. PRx utilizes the conserved catalytic cysteine residue, known as peroxidatic Cys (C_P_), to reduce the peroxides. Upon oxidation by peroxides, it yields cysteine sulfenic acid (CP-SOH), which eventually reacts with another cysteine residue denoted as resolving Cys (C_R_). The resultant disulfide bond C_P_-S-S-C_R_ is reduced by a suitable electron donor such as thioredoxin or glutaredoxin [6,46,47,48,49,50,51,52,53,54].

In addition to the removal of the ROS generated as a consequence of metabolic processes, there exists enzymes that are capable of alleviating the release of mt-ROS. The most effective mechanism for undertaking this is to ensure an efficient electron transport to oxygen, which acts as the terminal electron acceptor. In the presence of their substrates, enzymes like mitochondrial hexokinase and creatine kinase play a role in recycling ADP, thereby facilitating the efficient transfer of electrons to complex IV and sustaining ATP synthase activity. This also results in a low membrane potential [46,55].

An oxidase is present at the end of the mitochondrial ETC in higher plants known as alternative oxidase (AOX), which inhibits the accumulation of mt-ROS, aiding in better survival under stressful conditions [56,57]. It has been shown that salicylhydroxamic acid (SHAM), an inhibitor of AOX, has the capability to increase cellular ROS in *Pisum sativum* [58].

Another group of enzymes called sirtuins (SIRTs), nicotinamide adenine dinucleotide (NAD)^+^-dependent histone deacetylases, are capable of regulating ROS. In mammals, seven SIRTs, designated as SIRT 1–7, have been identified. Under stress conditions, various SIRTs are activated by NAD^+^. The activated SIRTs regulate the antioxidant response element, which eventually helps to maintain the redox signaling. SIRT 3, 4, and 5 are localized in the mitochondria. SIRT3 elevates the capability of MnSOD to reduce ROS, along with modulating the enzymes of OXPHOS. However, the role of SIRT 4 is not clear, though it has been reported that the overexpression of SIRT 4 increases ROS production, possibly by inhibiting the binding of Mn-SOD with SIRT 3. SIRT 5, on the other hand, is able to convert ammonia, which potentially may give rise to ROS, into nontoxic urea, thus regulating ROS indirectly [59,60].

It is a well-established fact that ROS oxidizes the biopolymer and weakens the proper functioning of physiological processes in an organism with age. With age, the effectiveness of the antioxidant system also reduces. To combat such a situation, researchers have developed a variety of ROS inhibitors. These inhibitors help not only in a therapeutic way but also aid in a better understanding of cellular functioning by utilizing them in in vitro studies. One of the mitochondrial-targeting antioxidants used lipophilic alkyl triphenylphosphonium cations as a conjugate, where the ubiquinone moiety is linked to the triphenylphosphonium cation by a C10 aliphatic chain and is known as MitoQ. Professor Vladimir Skulachev and his team replaced the ubiquinone with plastoquinone and called the resulting substance SkQ. It was reported that the SkQ derivative SkQ1, at lower concentrations, was able to prolong the life of organisms like *Drosophila* and mice. Reduced SkQ1 (SkQ1H2) was also able to cross microheterogenous lipid–water system, making it a better antioxidant, and to help with the mitigation of oxidative stress, cellular damage, and senescence [61]. Pretreating rats with SkQ1 reduced H_2_O_2_ levels and decreased the arrhythmia induced by ischemia in isolated hearts, prevented morbidity in kidney ischemia, and reduced the damage in myocardial infarction [62]. Another mitochondrial-targeted antioxidant is (2-(2,2,6,6-tetramethylpiperidin-1-oxyl-4-ylamino)-2-oxoethyl) triphenylphosphonium chloride monohydrate (mito-TEMPO). mito-TEMPO has shown promising results in diabetic cardiomyopathy. It not only reduced glucose-induced superoxide formation, but also reduced cardiomyopathic changes in both type 1 and type 2 diabetic mice [63].

The synthesis of ATP through complex V is coupled to the proton motive force, which, in turn, is derived from the redox reactions of the ETC [64]. However, under physiological conditions, OXPHOS is not always perfectly coupled and a few protons may leak, decreasing the proton motive force as well as the mitochondrial membrane potential, and resulting in reduced ROS production [65]. The proton leak could either take place via the basal or the constitutive proton leak or in a regulated or induced manner. However, the basal proton leak is not regulated but is dependent on the fatty acyl composition of the phospholipids of the IMM, and the basal conductance is modulated via the IMM protein adenine nucleotide translocase (ANT). Inducible proton leakage is governed by specific IMM proteins, like uncoupling proteins (UCPs). Five different types of UCPs have been identified in mammals, and are designated as UCP1-5. UCP1, which is mostly expressed in the mitochondria of the brown adipose tissue, regulates thermogenesis, whereas UCP2 and UCP3 are involved in mitigating mt-ROS levels. The correlation between proton leak and ROS generation is not straightforward, as studies have shown that, while uncoupling decreases ROS levels in certain scenarios, it increases ROS production in others. It has also been suggested that a protective feedback loop exists, as an increased ROS level activates the leakage of protons, subsequently resulting in decreased ROS production. However, it has also been shown that elevated mt-ROS levels are crucial for UCP1-mediated thermogenesis [66].

## 3. Signaling through Mitochondrial Reactive Oxygen Species

From mammals to plants, mt-ROS have been reported to play a crucial role in signaling processes. mt-ROS, like any other cell signaling event, turns on its signal transduction pathway in the presence of an external stimuli. This is evident from the adaptive response initiated during hypoxia(Figure 3). The increased mt-ROS generation at complex III is essential for the activation of hypoxia-inducible factor-1 (HIF-1) during hypoxic conditions (0.3–5% of O_2_) to regulate cellular responses. When O_2_ levels fall below 0.3%, this leads to decreased levels of ATP, thus activating adenosine monophosphate (AMP)-activated protein kinase (AMPK) to replenish the ATP levels. Interestingly, it has been reported that, during hypoxic conditions, mt-ROS activates AMPK, clearly suggesting the role of mt-ROS in the AMPK-mediated recovery of ATP levels. However, considering mt-ROS as a non-canonical pathway, on activation, AMPK triggers antioxidant PGC-1α activation, which acts as a feedback mechanism for mt-ROS control. Also, a mitochondrially localized AMPK can detect unhealthy mitochondria due to stress and can mediate mitophagy, utilizing ULK1 kinase [67]. It has also been reported that mt-ROS aid in the epithelial–mesenchymal transition (EMT) in alveolar tissues under hypoxic conditions [68,69,70,71]. mt-ROS is also implicated in the inhibition of adipocyte differentiation by positively regulating the adipogenic repressor CHOP-10/GADD153 under hypoxic conditions [72].

Apart from aiding in adaptation under hypoxic conditions, mt-ROS is also associated with the regulation of transcription factors. For instance, NF-E2-related factor 2 (Nrf2) is a transcription factor regulated by mt-ROS to maintain mitochondrial homeostasis via various protein kinases. Usually under physiological conditions, it maintains the mitochondrial redox homeostasis by activating the antioxidant defense machinery and facilitates mitochondrial biogenesis. However, under severe stress condition, Nrf-2 may induce Klf9 expression, which inhibits mt-ROS-scavenging enzymes and promotes apoptotic cell death(Figure 3). This intricate relationship between mt-ROS and Nrf2 has been documented in a recent review article by Kasai et al. [73].

mt-ROS also initiates mitochondrial retrograde regulation in *Arabidopsis* by utilizing the ANAC017 pathway, associated with the ER [74]. mt-ROS also play a crucial role in promoting the functions of plant hormones. For example, mt-ROS act as a secondary messenger for abscisic acid (ABA), which functions in seed dormancy and germination and in stomatal movement in *Arabidopsis thaliana*. It has been reported that ABA overly sensitive 6 (abo6) shows higher levels of mt-ROS. This accumulation of mt-ROS is aided by DEXH box RNA helicase, a subfamily of DEAD box proteins in *A.thaliana* encoded by abo6. DEXH box RNA helicase regulates the splicing of various genes of complex I. It has been reported that, in *Arabidopsis thaliana*, accumulated mt-ROS, specifically H_2_O_2_, aids in stomatal closure. Also, the supplementation of mitochondria-specific antioxidant Mito-Q, reduced GSH, and auxin were able to mitigate the seed dormancy and ceased seed germination and primary root growth, induced by ABA [75,76].

MAO-A-induced ROS and mt-ROS aid in protective cellular responses, such as autophagy, as evident from human SH-SY5Y neuroblastoma cells. SH-SY5Y neuroblastoma cells have a higher MAO-A protein expression but the activity is modestly increased. The associated enhanced ROS levels in these cells showed higher cellular protein oxidation. This may lead to cellular damage and, consequently, apoptosis. In order to alleviate apoptosis, there is a clearance of the damaged proteins and organelles via autophagy; more specifically, mitophagy. The oxidation of proteins induces the lysine-63 linked ubiquitination of mitochondrial protein, which triggers autophagy. mt-ROS additionally trigger the phosphorylation of dynamin-1-like protein, consequentially aiding the fragmentation of mitochondria and its clearance. MAO-A-induced mt-ROS are also implicated in heart failure. Chronic postischemic remodeling includes various changes, such as interstitial fibrosis, cardiomyocyte death, and mitochondrial dysfunction leading to heart failure. Of these, mitochondrial dysfunction has crucial mediators known as 4-hydroxynonenal (4-HNE). With the help of adult mice cardiomyocytes, neonatal rat ventricular monocytes, and transgenic mice overexpressing MAO-A, it was shown that the activation of MOA-A, either with the help of a monoamine compound—tyramine—or in transgenic mice hearts, elevated the levels of H_2_O_2_. It was also established that MOA-A-induced mt-ROS led to cardiolipin peroxidation, via hydroxyoctadecadienoic acids (HODEs) synthesis as an intermediate, and led to the intramitochondrial accumulation of 4-HNE.This 4-HNE binds to a voltage-dependent anion channel (VDAC) and a mitochondrial calcium uniporter. This leads to Ca^2+^ accumulation, mitochondrial dysfunction, a loss of membrane potential, and, consequentially, a reduced ATP level [43,77].

In *C. elegans* ETC mutants, *isp-1* and *nuo-6*, increased mt-ROS induces an intrinsic apoptosis pathway (composed of CED-9, CED-4, and CED-3) mediated by CED-13 protein. This pathway reduces ATP levels and utilizes it for protective functions. In wild-type worms, this pathway provides protection against mitochondrial dysfunction, whereas the prolonged activation of this pathway aids in the longevity of the mutants, independent of apoptosis [78]. On the contrary, with the help of the *Saccharomyces cerevisiae* mutants GS129 and GS130, which have a mutation in the amino terminal domain (ATD) of mitochondrial RNA polymerase, it has been shown that defective mitochondrial gene expression leads to the faulty assembly of the OXPHOS system. This leads to an increased level of mt-ROS, conditional impairment of the respiratory chain, and a consequential reduction in lifespan. It was established that all mutations and increased mt-ROS levels do not hamper the lifespan until and unless the mt-ROS levels are elevated beyond a threshold point. For instance, in GS129 mutants with a *rpo*41-R129D mutation, the increase in mt-ROS was high, and it aided in both the impairment of the respiratory chain and a subsequent reduction in lifespan, as the antioxidant enzyme SOD was able to rescue these damages. However, in GS130 mutants with a mutation in *rpo*41-D152A/D154A, the accumulation of ROS was intermediate, and SOD was not able to rescue the mild reduction in lifespan [79]. As mentioned earlier, ROS is also generated at complex I, and this ROS has the potential to oxidize cysteine residues to initiate downstream signaling. Cysteine thiol groups are highly reactive in their deprotonated state, which is favored by the higher pH of the mitochondrial matrix. The deprotonated thiolate anions act as redox switches for signaling and are oxidized to sulfenic acid intermediates by ROS, modulating the effector protein’s structure and function. H_2_O_2_ oxidizes them to sulfenic acid, whereas superoxide anion oxidizes them to thiyl radicals [80]. Few phosphatases, like PTP1b and PTEN, have cysteine residues, which could be modulated by H_2_O_2_. The oxidation of cysteine residues mediated by H_2_O_2_ also plays a crucial role in protein folding in mitochondrial intermembrane space via the mitochondrial import and assembly (MIA) pathway. Proteins like MIA40-CHCHD4 and ERV1-ALR are involved in the MIA pathway, wherein cysteine residues are oxidized and the proteins are folded in their mature form. The cysteine residue of the MIA pathway shows a particular pattern, known as twin CX3C or CX9C motifs, which stabilize the protein structure and import by forming a disulfide bridge between these motifs. Hydrophobic membrane proteins are transported through the intermembrane space via the twin CX3C motif, whereas CX9C motifs are responsible for the biogenesis of cytochrome *c* oxidase (COX) and is mediated by ROS and redox-regulated steps. H_2_O_2_ oxidizes the Cys37 residue of the mitochondrial targeting sequence of COA8, which aids with its stability and import within the mitochondria. This ultimately helps in COX assembly [81].

Complex I-mediated ROS is crucial for the behavioral response under acute hypoxia. This is evident from *C. elegans.* Under acute hypoxia, *C. elegans* opts for certain behavior, which includes an increase in locomotory speed and turning angle. However, this was completely stopped in *C. elegans*, which had a mutation in the ortholog of the complex I subunit NDUFS2, known as *gas-1*. This mutation also increased ROS; in order to confirm the change in avoidance behavior, a Mn-SOD mutant was used, which showed partial inhibition of the behavioral response to acute hypoxia, whereas mutations in all SOD genes resulted in the complete inhibition of the acute hypoxia response [80].

mt-ROS plays a crucial role in Ca^2+^ signaling in the cells as well. The activation of inositol 1,4,5-trisphosphate (IP3) and ryanodine receptors triggers calcium (Ca^2+^) release and mobility, which induces mt-ROS generation. This elevated ROS level further amplifies Ca^2+^ signaling and supports specific cellular functions. However, it has also been reported that, in the endothelium of rat mesenteric arteries, H_2_O_2_ attenuates the uptake of Ca^2+^ by mitochondria and inhibits the IP3 receptor-induced release of Ca^2+^. This aids in reducing the excess calcium load on mitochondria via the depolarization of the membrane potential [82,83]. Under hypoxic conditions, complex I of ETC tends to adopt a dormant state, which alters its redox properties by inducing a structural rearrangement in the I_Q_ site and contributing to electron leakage; this enhances superoxide anion production. These excess superoxides are dismutated to H_2_O_2_, which is the most preferred ROS for redox signaling. Elevated H_2_O_2_ levels trigger an increase in intracellular Ca^2+^ levels that initiates the release of neurotransmitters and other mediators in cells like astrocytes, glomus cells, and pulmonary artery smooth muscle cells, helping with physiological responses [80,81].

One of the most intriguing functions of mt-ROS lies in immune responses. Immune cells undergo metabolic changes upon activation in order to meet energy requirements. Mitochondria play a key role in these metabolic adaptations as they contain the enzymes essential for the tricarboxylic acid (TCA) cycle and cellular respiration. A higher amount of mt-ROS is produced, which is associated with these metabolic changes. Ca^2+^ release is triggered by T cell receptors (TCR). This results in the uptake of Ca^2+^ by mitochondria and, consequentially, the activation of TCA cycle enzymes, aiding mitochondrial respiration and providing energy compensation for activated T cells. With mitochondrial respiration, mt-ROS production is induced and aids in T cell activation. For instance, mt-ROS is required for the induction of interleukin-2 (IL-2) via the activation of the nuclear factor of activated T cells (NFAT). This has been established experimentally by using Rieske iron–sulfur protein (RISP), a complex III component knocked out in mice. The CD4 cells lacking RISP showed reduced mt-ROS levels and an impaired IL-2 induction. Additionally, it was shown that, in RISP knockout, the absence of complex III mt-ROS was compensated by the addition of exogenous H_2_O_2_. The mechanism proposed by the researchers is that the stimulation of TCR leads to the activation of the calcium release-activated calcium (CRAC) channel, which induces calcium influx and results in mt-ROS production, which ultimately aids in T cell activation and IL-2 production [84,85,86]. Another study showed that, during T cell activation, mt-ROS is produced via glycerol-3-phosphate dehydrogenase (GPD2). In activated T cells, a shift towards glycolysis is preferred for energy demand. During this process, glycerol-3-phosphate (G3P) is produced and taken up by the mitochondria. In mitochondria, G3P is converted to dihydroxyacetone phosphate (DHAP) by GPD2. This activation of GPD2 leads to ubiquinol accumulation and ultimately mt-ROS generation, which aids in nuclear factor κB (NF-κB) activation and the release of ILs [87]. It has also been shown that macrophages activation is aided by mt-ROS to elicit an effective antimicrobial response [88]. Therefore, it is quite evident that mt-ROS plays a crucial role in the working of the immune system.

## 4. Mitochondrial ROS in Immune Signaling

As we have seen in the previous section, mt-ROS is involved in major immunological events like T cell activation. Here, we will try to give an idea about how mt-ROS is crucial in immune responses.

Toll-like receptors (TLRs) are considered to be crucial in innate immunity, the first line of defense against pathogenic infections. There exist 10 different members of TLRs, of which TLR-1, -2, -4, -5, -6, and -10 are found on the surface of the innate immune cells, while TLR-3, -7, -8, and -9 are present as intracellular pattern recognition receptors (PRRs) on different organellar membranes. Each TLR recognizes unique pathogen-associated molecular patterns (PAMPs). TLR signaling ultimately activates the NF-κB transcription factor and mitogen-activated protein kinase (MAPK) pathways, which initiate the secretion of proinflammatory cytokines [89,90]. It was reported that TLR1/2/4 signaling, upon recognizing bacteria-associated PAMPs, recruits one of the adaptor molecules, tumor necrosis factor receptor-associated factor 6 (TRAF6), in the mitochondria, where it interacts with *e*volutionarily *c*onserved signaling *i*ntermediate in *T*oll pathways (ECSIT). ECSIT has been implicated in complex I assembly. The interaction leads to the ubiquitination of ECSIT and its accumulation at the OMM periphery, causing increased levels of mt-ROS production. This protein–protein interaction also recruits mitochondria to the phagosomes, enhancing the bactericidal activity. In the absence of TRAF6 or ECSIT, the bactericidal activity of macrophages is significantly reduced, as evident from the ECSIT- and TRAF6-deficient macrophages. When bone marrow-derived macrophages (BMDM) were infected with *Salmonella typhimurium*, a decreased level of mt-ROS was observed, which led to higher levels of bacteria. In fact, the mitochondria-specific expression of the antioxidant enzyme catalase also reduces the antibacterial immune response of the macrophages, clearly showing that mt-ROS plays a significant role in this response. This TRAF6/ECSIT complex formation is aided by Mst-1 and Mst-2 kinase by activating GTPase RAC, which ultimately triggers TLR signaling for ROS generation [91,92]. Neutrophils are another kind of immune cells which induce a variety of defense mechanisms at the site of infection, including phagocytosis, degranulation, and the release of neutrophil extracellular traps (NETs), a process called NETosis. mt-ROS is crucial for neutrophil activation and inducing these effects. It has been observed that treating N-formyl-methionyl-leucyl-phenylalanine (fMLP)-induced neutrophils with mitochondrial-specific antioxidant (SkQ1) downregulates mt-ROS levels and, as a consequence, decreases the exocytosis of azurophils and specific granules. Additionally, it inhibited the oxidative burst induced by NOX2. However, this reduced level of mt-ROS aided in the apoptosis of fMLP-induced neutrophils. mt-ROS also aids in the formation of NETs in Ca^2+^ ionophore A23187-induced neutrophils. In chronic granulomatous disease (CGD), a genetic condition with impaired NADPH oxidase, mt-ROS aids in NETosis, as evident from the observation that treating CGD neutrophils with SkQ1 inhibited A23187-induced NET formation [93,94].

It has been reported that, in *Listeria monocytogenes*-infected macrophages, TRAF-6 induced mt-ROS is essential for the activation of NF-κB essential modulator (NEMO), a subunit of Iκ-B kinase. mt-ROS aids in the intermolecular linkage of NEMO via a disulfide bond between cysteines that are redox-sensitive: Cys54 and Cys347. This leads to the phosphorylation of Iκ-B and releases NF-κB to translocate to the nucleus and regulate the transcription of immune mediators [89,95]. Additionally, SkQ1 has the ability to inhibit TNF-induced NF-κB activation, which, in turn, suppresses the release of proinflammatory cytokines and the expression of cell adhesion molecules (CAM) in human endothelial cells and reduces neutrophil adherence, thus establishing the role of mt-ROS in age-associated vascular inflammation [96]. Treating L6 myotubes with a NOD-2 activator, muramyl dipeptide (MDP), led to oxidative stress and mitochondrial dysfunction, which was evident from higher mt-ROS production, reduced ATP levels, modulated membrane potential, and alterations in the expression of genes related to mitochondrial activity and metabolism. This mitochondrial dysfunction and oxidative stress led to insulin resistance. It has also been shown that NOD-2 initiates an autonomous immune response which aids in the development of insulin resistance [97,98].

mt-ROS also plays a crucial role in the activation of the NOD-like receptor (NLR) family pyrin domain containing 3 (NLRP3) inflammasome (Figure 4). Inflammasomes are multiprotein complexes that reside in the cytoplasm and play important functions as innate immune mediators. NLRP3 is an intracellular sensor which, on interaction with apoptosis-associated speck-like protein containing a caspase recruitment domain (ASC) protein, forms a caspase-1-activating NLRP3 inflammasome that is considered crucial for pathogen clearance, inflammation, and pyroptosis, by secreting the matured pro-inflammatory cytokine IL-1β. The expression of NLRP3 is dominant in neutrophils and macrophages with lower levels of expression in nonimmune cells. When acted upon by an external stimulus, a single unit of NLRP3 pyrin domain interacts with the ASC pyrin domain, and ASC interacts with pro-caspase-1 with a CARD domain, forming the inflammasome (Figure 4). This interaction releases active caspase-1, which in turn regulates the expression of inflammatory cytokines. In general, the protein levels for NLRP3 are low and, in order to mediate a successful inflammasome formation, both transcriptional and translational level signaling is regulated. This includes TLR-mediated NF-κB activation, which, as stated earlier, is aided by mt-ROS and post-translational modifications. It has been established that a deficiency of autophagic protein beclin-1 leads to mt-ROS generation and activation of the NLRP3 inflammasome. Another mechanism which has been proposed but not understood well is NLRP3 activation by thioredoxin-interacting protein (TXNIP). TXNIP binds to and negatively regulates the expression of antioxidant protein thioredoxin-1 (Trx1). However, increased ROS levels lead to the disassociation of TXNIP from Trx1 and facilitates the interaction between TXNIP and NLRP3, activating NLRP3. Additionally, mt-ROS-targeting antioxidants inhibit NLRP3 activation [99,100,101]. All these observations collectively establish a crucial role of mt-ROS in the activation of innate immune effector responses.

## 5. Mitochondrial ROS in Bacterial Infection

Pathogenic bacteria employ diverse mechanisms within hosts to enhance their virulence, while hosts employ various mechanisms to eliminate these pathogens. Mitochondrial reactive oxygen species (mt-ROS) play a pivotal role in this interplay (Table 1).

In *C. elegans*, it has been reported that elevated levels of mt-ROS via HIF-1/AMPK helps in attaining immunity against pathogenic *Enterococcus faecalis*. However, it has also been reported that a mutation in the iron–sulfur protein of *C. elegans* leads to toxicity when in association with an enterobactin–iron complex, which increases mt-ROS [105,120].

To evade the host immune system, *L. monocytogenes* utilizes NLRX1, the unique member of NLR with a mitochondrial targeting sequence. NLRX1 associates with LC3 (microtubule-associated proteins 1A/1B light chain 3B) to induce mitophagy, regulate membrane potential, and suppress mt-ROS generation. This was suggested in experiments using NLRX1-deficient macrophages in vitro and in vivo, where, in the absence of NLRX-1, mt-ROS generation was increased, boosting the bactericidal abilities of the host cell. However, treatment with mito-TEMPO, a ROS scavenger, alleviated and suppressed the bactericidal effects. Further probing revealed that the pathogen uses its virulence factor listeriolysin O (LLO) to oligomerize NLRX1, which facilitates the binding of LC3-interacting region (LIR) of NLRX1 to LC3, leading to the induction of mitophagy in host cells. The attenuation of NLRX1 or administration of mitophagy inhibitors increased the generation of mt-ROS, resulting in the suppression of *L. monocytogenes* survival. This shows that *L. monocytogenes* exploits NLRX1 to induce mitophagy to evade the host immune response [121]. In another study, murine bone marrow-derived dendritic cells were infected with *L. monocytogenes* to analyze the membrane proteins. It showed that dynein light chain 1 (Dynll1), along with the mitochondrial cytochrome oxidase Cox4i1, is crucial for bactericidal effects. Upon infection with *L. monocytogenes*, the Dynll1-Cox4i1complex is disrupted and aids in the release of mt-ROS. This mechanism is proposed from the results of experiments performed with different mutants. The phagosome-rupturing enzyme LLO-deficient strain of *L. monocytogenes* was not affected in Dynll1-deficient cells; however, the titer of the lysosomal-escaped bacteria was low in Dynll1-deficient cells. Hence, it could be concluded that Dynll1 responds to intracellular proliferation and that its deficiency inhibits the intracellular proliferation of the pathogen. Although Cox4i1-deficient cells showed a lower oxidative burst, the knockout of Dynll1 enhanced the strength and duration of the oxidative burst. Furthermore, treatment of sodium azide, an inhibitor of cytochrome c oxidase, in Dynll1-deficient cells increased the bacterial titer. In a double knockout of Dynll1 and Cox4i1 cells, the bacterial titer was significantly low. These results establish that, upon dissociation of Dynll1, Cox4i1 aids in mt-ROS production and facilitates the bactericidal effects. However, alternate mechanisms do exist, where, in the absence of Dynll1, Cox4i1 can clear the bacterial infection, albeit with a lower efficiency than the combined effects of the Dynll1-Cox4i1 complex [122]. Additionally, infecting HeLa cells with different *L. monocytogenes* mutants, including mutants lacking hemolytic activity of LLO (LmΔhly) and phospholipase activities of phosphatidylinositol (PI) and phosphocholine (PC)—Lm ΔPI-PLC and Lm Δ PC-PLC, respectively—revealed some interesting insights regarding the correlation between the infectivity and mt-ROS. It was observed that cells infected with Lm Δ PC-PLC showed higher mt-ROS generation compared to Lm Δ PI-PLC, Lm Δ hly, and wild-type cells. Furthermore, the one infected with Lm Δ hly showed the lowest levels of mt-ROS. Thus, explaining the importance of PC-PLC in reducing the mt-ROS levels of a host induced by LLO [123].

In response to PAMS, the host’s respiratory chain is modulated, which leads to increased mt-ROS levels, the activation of caspase 1, and the release of antimicrobial metabolites to clear the bacterial invasion. *Staphylococcus aureus*, a Gram-positive bacterium infection, induces NLRP3 inflammasome activation. The activated NLRP3 activates caspase-1 and interleukins, which, on the one hand, have been shown to aid *S. aureus* virulence, evidenced by the attained protection against *S. aureus* in NLRP3-deficient mice. On the other hand, it has been reported that caspase 1 activation is required for *S. aureus* killing. Regarding the virulence factor of *S. aureus*, the alpha toxin (AT) plays a crucial role in the NLRP3-mediated impairment of the host immune system. It was established that the alpha toxin-induced activation of the NLRP3 inflammasome in *S. aureus* leads to the sequestering of caspase-1 away from the phagosome. This was evidenced by inhibiting NLRP3 and ASC by administering siRNA in BMDMs. It was observed that, in these BMDMs, caspase-1 was more localized, and the killing was improved. Also, in AT-deficient mutant *S. aureus* infections, mt-ROS was significantly increased in human monocytes, emphasizing the fact that AT attenuates mt-ROS activity. In order to answer the underlying mechanism, live imaging of the infected monocytes was performed, which showed a high association of mitochondria with the internalized bacteria in AT-deficient strains as compared to wild-type cells. Hence, it was concluded that AT of *S. aureus* leads to the dissociation of mitochondria and the impairment of the defense responses associated with it. It was also shown that, in the absence of AT, ETC complex II activity was able to kill the bacteria effectively [124]. In neutrophils, NOX2 is considered to be a major source of ROS rather than mitochondria. However, it has been shown that treating neutrophils with ETC inhibitors, specifically the one for complex III, reduces mt-ROS production, NET formation, and bactericidal activity against *S. aureus* [125]. *S. aureus* that are resistant to a broad range of β-lactam antibiotics are known as MRSA (methicillin-resistant *Staphylococcus aureus*). After the engulfment of MRSA by macrophages, there is an elevated level of ER stress, which leads to mt-ROS generation, which is then converted to H_2_O_2_ by Mn-SOD. To deliver this mt-H_2_O_2_ to the phagosome, mitochondrial-derived vacuoles (MDVs) are formed. The MDVs containing mt-H_2_O_2_ are induced by TLR signaling and the Parkin/Pink1-dependent mitochondrial stress pathway. All these conclusions were evidenced with the help of both in vitro and in vivo experiments. MRSA, being antibiotic-resistant, has the capability to reside in macrophages without being killed. This is attained with the help of caspase-11, as it restricts the recruitment of mitochondria in the vicinity of the bacteria-containing vacuole. This is perceptible by the observation wherein caspase-11 deficiency aided in the better association of mitochondria with bacteria-containing vacuoles and enhanced mt-ROS. Additionally, in caspase-11-deficient macrophages, treatment with antimycin-A increased mt-ROS generation and, consequently, led to better bactericidal effects [126,127].

Tumor necrosis factor (TNF) is a critical immune modulator in the context of *Mycobacterium tuberculosis* infection. Zebrafish with high levels of proinflammatory *LTA4H* (leukotriene A4 hydrolase) presented excess TNF production. It has been observed that, with an initial bactericidal effect, elevated TNF levels lead to the necrosis of host macrophages, resulting in the release and proliferation of the pathogen in the extracellular environment. It has been established that, under both conditions, mt-ROS plays a crucial role. Enhanced levels of mt-ROS are generated via the receptor-interacting serine-threonine kinases 1 and 3 (RIP1 and RIP3)-mediated pathway. The RIP1-RIP3 pathway is well established in TNF-mediated programmed necrosis. With the utilization of its substrate, PGAM5, enhanced production of mt-ROS is triggered, which leads to host cell necrosis and the release of bacteria in the extracellular environment for proliferation. The enhanced mt-ROS initially aids in bactericidal activity but later on it activates cyclophilin D (CYPD) and acid sphingomyelinase (aSMAse). CYPD aids the mitochondrial permeability transition pore formation and aSMAse aids in the formation of ceremide; both of these processes lead to cell death. In agreement, the inhibition of CYPD and aSMAse decreased the necrosis of macrophages and enhanced bactericidal activity [128,129]. Another mechanism established for the TNF-induced enhancement of mt-ROS in zebrafish and human macrophages includes RET. Excess TNF causes an increased glutamine uptake and increased succinate accumulation. Oxidation of the increased succinate by complex II drives the RET and generates mt-ROS at complex I. This is evident as utilizing metformin, a complex I inhibitor, reduces mt-ROS levels and inhibits the necrosis of *M. tuberculosis*-infected macrophages [130]. To survive within the macrophages, *M. tuberculosis* exploits host metabolic pathways and uses its nutrients. It has been observed that fatty acid oxidation (FAO) is one such crucial pathway, as the inhibition of FAO suppresses the survival of *M. tuberculosis* in mice macrophage cells. The inhibition of FAO also increases mt-ROS generation, which in turn triggers NADPH oxidase and xenophagy to combat *M. tuberculosis* infection [131].

*S. typhimurium* effector protein SopB prevents mt-ROS generation by interacting with TRAF-6, thereby inhibiting host cell apoptosis and facilitating *S*. *typhimurium* proliferation [132]. It has been reported that treating human macrophages with histone deacetylase inhibitors during infection increases mt-ROS and helps in the clearance of Gram-negative *Salmonella typhimurium* [133]. Mouse model and mammary epithelial cells (MECs) infected with *Streptococcus uberis* induced mt-ROS via TLR2 signaling. The TLR2-deficient mice showed severe pathological conditions [134].

*Pseudomonas aeruginosa* utilizes pyocyanin, a membrane-permeable toxin, in order to escape the host immune response. Pyocyanin induces mt-ROS generation by interacting with the mitochondrial respiratory chain proteins. This activates the mitochondrial acid sphingomyelinase, the synthesis of ceramide, and the release of cytochrome, ultimately culminating in neutrophil death [106]. This observation clearly suggests that pyocyanin-induced neutrophil death is mediated by elevated levels of mt-ROS. In *C. elegans*, elevated levels of mt-ROS are implicated in its defense against *P. aeruginosa* infection [105].

## 6. Mitochondrial ROS in Protozoan Infection

Protozoan parasites are unicellular eukaryotes, which pose a major threat to human and livestock health across the tropical regions of the globe. These parasites not only contribute to higher morbidity rates but also lead to economic challenges. While they majorly burden tropical and subtropical regions, some temperate regions are also affected. These infectious parasites include kinetoplastids, like the Trypanosoma and Leishmania species; apicomplexan parasites, like Toxoplasma and Plasmodium; diplomonadids, like the Giardia species; amoebozoans, including Entamoeba histolytica; and axostylata, like the Trichomonas species. Chronic protozoal infections lead to tissue damage, which often is a consequence of an immune response or cytokine profile modulation. However, there is no significant vaccine against these infections and the available drugs do not suffice, as there is a higher propensity of drug resistance. Understanding the role and regulation of mt-ROS can alleviate these issues [135,136,137]. *Trypanosoma cruzi*, the causative agent of Chagas’ disease, leads to cardiomyopathy as a major consequence. When cardiomyocytes are infected with *T. cruzi*, there is an impairment in ETC and an elevation in ROS production. However, no other enzymes, such as NADPH oxidase or xanthine oxidase activity, were increased, indicating that mitochondrial dysfunction plays a crucial role in the *T. cruzi*-induced defense mechanism and is a potential source of ROS generation in this process [138]. Meanwhile, in the host, mt-ROS aids in parasitic control; the mt-ROS formed in *T. cruzi* helps in their proliferation (Table 1) by regulating the redox balance, which possibly aids with their adaptation to the host environment. *T. cruzi*, as a kinetoplastid, consists of a single mitochondrion. It has been reported that exposure of the heme group (a common scenario in the midgut of the insect vector) does not affect the ultrastructure of the parasite; however, it modulates mitochondrial physiology and triggers the enhanced generation of mt-ROS. Enhanced levels of mt-ROS are also responsible for the hyperpolarization of the mitochondrial membrane potential. Treating the parasites with an uncoupler and mt-ROS scavengers such as carbonyl cyanide p-(trifluoromethoxy) phenylhydrazone (FCCP) and MitoTEMPO reduced mt-ROS and ATP levels, which consequently reduced cell proliferation significantly. Therefore, heme-mediated mt-ROS can regulate ATP levels by modulating mitochondrial bioenergetics and can thus contribute to parasitic proliferation by helping with adaptation to the host environment [109]. A study, where CD4 T cells were evaluated in the non-infected, acute, and chronic phases of *T. cruzi*-infected mice, showed that CD4^+^ T- cells had an elevated proton leak and UCP3 upregulation, which led to mt-ROS accumulation and the depolarization of the mitochondrial membrane, which could contribute to an immunocompromised condition during acute infection. These cells were more prone to apoptosis and had a metabolic shift towards glycolysis. Additionally, it was observed that the expression of Mn-SOD increased; however, only after treating the cells with antioxidant N-Acetyl Cysteine (NAC) was the apoptosis of CD4T cells prevented [139]. A significant role of the mTOR (mammalian target of rapamycin) pathway has been implicated in *T. cruzi* survival in infected macrophages, as inhibiting mTOR reduced the *T. cruzi* load. It has been shown that treating the BMDM with a rapamycin inhibitor of the mTOR pathway increased the production of pro-inflammatory cytokines and reduced anti-inflammatory cytokines. Also, enhanced mt-ROS generation along with NLRP3 expression was observed to be crucial for *T. cruzi* proliferation control. Hence, it can be inferred that a potential association exists between mTOR inhibition, mt-ROS generation, and NLRP3 formation, as, when attenuating mt-ROS using MitoSOX and NLRP3 knockout mice, the effect of rapamycin was partially reversed. The inhibition of mt-ROS aided in parasitic replication, and there was an increase in the parasitic load in NLRP3 knockout macrophages pretreated with rapamycin [140].

*Toxoplasma gondii* is another parasite which can be fatal and causes toxoplasmosis, affecting 30% of the world’s population. During active invasions by parasite, elevated levels of mt-ROS have been reported, with reduced expressions of OXPHOS protein, indicating mitochondrial dysfunction (Table 1). Upon infection with *T. gondii*, the P2x7 receptor of macrophages is activated by extracellular ATP; eventually, it activates NLRP3 inflammasome formation via caspase 1, and this triggers IL-1β secretion and an elevation in mt-ROS production, leading to subsequent antiparasitic activity. It was established that the inflammasome formed was canonical in nature by utilizing knockout mice for the different components of inflammasome [110,141]. It has been shown that MitoTEMPO, a mt-ROS scavenger, to some extent, could reduce the proliferation of *T. gondii*. However, the major defense was via mitochondrial fusion, so that the parasites are unable to utilize the host’s fatty acid [142]. It has been reported that, in *T. gondii*, a cytosolic catalase exists that is able to cleave H_2_O_2_ and aids in inducing virulence in mice, hence indicating that *T. gondii* targets the ROS of the host to induce virulence [143]. As mentioned earlier, UCP2 is involved in reducing mt-ROS levels. It is also involved in immune response suppression during *T. gondii* infection, as evidenced by the research carried out on UCP2-deficient mice. These mice, upon infection with *T. gondii*, showed complete resistance. There was a significant reduction in parasitic cysts and inflammatory sites in the brains of these mice, and the macrophages were able to produce more ROS than the wild-type and aided in eliminating parasites. However, the exposure of isolated macrophages from the mutant mice to a ROS quencher (L-histidine) reduced their parasiticidal effects [144].

*Leishmania*, the parasite that causes leishmaniasis, has various species, and the severity of the infection depends on the species of *Leishmania* causing the disease [145]. *L. donovani*, the causative agent of visceral leishmaniasis, induces mitochondrial uncoupling protein 2 (UCP2) to downregulate mt-ROS levels and to successfully proliferate inside host macrophages [113]. However, the knockdown of macrophage UCP2 results in a lower parasitic load and increased levels of mt-ROS [114] (Table 1). UCP2, along with A-20, a NF-κB inhibitor, deactivates NLRP3 inflammasome formation and enhances disease propagation [146]. ROS generation in the *Leishmania*-infected macrophages is suppressed by upregulating heme oxygenase-1 (HO-1), which in turn attenuates the heme-dependent maturation and assembly of NADPH oxidase by downregulating gp91phox [147]. Mitochondrial dysfunction in infected macrophages modulates the microRNA dynamics; the stability of microRNA ribonucleoprotein complexes is increased, which reduces proinflammatory cytokine generation, destabilized ER-mitochondrial dynamics and, hence, promotes the survival of *L. donovani* [148]. Another species of *L. infantum*, infecting both humans as well as zoonotic host dogs, can concur with *Borrelia burgdorferi*. When coinfected with *B. burgdorferi*, ROS and mt-ROS production is reduced in *L. infantum*-infected macrophages. It has been observed that Nox-2 expression is reduced significantly. Additionally, Mn-SOD is upregulated, which could be a possible mechanism for the decrease in mt-ROS. It has been explained that, through the regulation of mt-ROS in the matrix, *B. burgdorferi* can alleviate cellular death and can aid *L. infantum* to thrive. However, this coinfection also leads to the release of proinflammatory cytokines [149,150].

The intestinal protozoan *Giardia duodenalis* hampers the growth of epithelial cells via an obscure mechanism. *Giardia* trophozoites were found to enhance ROS production when they infected *Caco-2* cells. Although it was not clarified that the generated ROS was mt-ROS, significant damage to the mitochondria was observed upon infecting the cells with *Giardia* trophozoites. This indicates that a close association exists between ROS and mitochondrial dysfunction in the parasitic-induced apoptotic pathway. It was shown that, upon treating the cells with trophozoites along with increased mt-ROS, the release of lactate dehydrogenase (LDH) was also escalated. This elevated release of LDH indicates membrane disruption and damage to the cell and could be a potential pathway for ROS release outside the cell to combat *Giardia* infection. Along with these changes, the modulation of the mitochondrial membrane potential, alterations in the expression levels of anti-apoptotic protein Bcl-2 and Bax protein, cytochrome c release, and the activation of key components of caspase-mediated apoptotic pathways were observed. However, pretreatment with the ROS inhibitor N-acetyl-cysteine (NAC) reversed *Giardia* trophozoite-induced apoptosis. Thus, it was concluded that, in Coca-2 cells, a ROS- and mitochondria-mediated caspase-dependent apoptotic pathway was induced upon *Giardia* trophozoite infection [151]. Similarly, Trichomonas vaginalis, the causative agent of *Trichomoniasis*, leads to the apoptosis of host cells. In vitro studies have shown that *T. vaginalis* can induce apoptosis in a wide variety of cells, including macrophages, vaginal epithelial cells, and human cervical cancer cells (SiHa cells). Quan et al. scrutinized the underlying mechanism which aids in the induction of apoptosis in the human cervical mucosal epithelial cancer cell line SiHa. They found that *T. vaginalis* triggers the generation of intracellular and mt-ROS upon infection in a parasite burden-dependent manner. This leads to the release of cytochrome c, a reduction in the expression of Bcl-2, and the activation of mitochondrial-dependent caspase-3. *T. vaginalis* was able to attenuate NF-κB p65 nuclear translocation and consequently downregulated NF-κB activity in a parasite load-dependent manner. All these events led to DNA fragmentation and resulted in apoptosis. However, NF-κB inactivation was reversed by pretreating the cells with the ROS scavenger NAC. This indicates that reduced NF-κB activation is associated with increasing intracellular and mt-ROS. Therefore, it can be concluded that *T. vaginalis* induces apoptosis in cervical mucosal epithelial cells through ROS generation, impacting NF-kB signaling and mitochondrial pathways, and that these could be potential targets for therapy [152].

## 7. Mitochondrial ROS in Viral Infection

Viruses are a class of obligate intracellular pathogens, which manipulate the molecular mechanisms of the host cell to accomplish their life cycle, including the replication of the genetic material (DNA/RNA), the translation of proteins and their modification, and the assembly and release of the viral particles (Table 1) [153]. Viruses have the capability to either tunicate their own structure or to hamper host antiviral defense, and mitochondria and mt-ROS both are implicated in the same. The mitochondrial antiviral-signaling protein (MAVS) is a major component for RLR-initiated signal transduction, which modulates the levels of type I interferon (IFN) as a part of the antiviral innate immune response [154]. It has been established previously that single-stranded RNA viruses are recognized by RLRs and initiate the production of type I IFN via the IFN-β promoter stimulator 1 (IPS-1), which is associated with mitochondria. It has been reported that autophagy is crucial for mediating this antiviral defense. In autophagy-defective cells that are Atg5 deficient, there was a higher level of RLR signaling, triggering more IFN secretion, which ultimately enhanced resistance towards infection by a vesicular stomatitis virus. This was mainly due to the accumulation of dysfunctional mitochondria, IPS-1, and ROS initiating a false RLR signal. Additionally, autophagy-independent cells triggered mt-ROS generation upon treatment with rotenone, inducing an enhanced level of signaling. However, antioxidants were able to curb these signals. This helps to establish that autophagy is a crucial process in antiviral defense, and that is why many viruses have evolved to block the host’s autophagy [155]. One such virus which negatively regulates autophagy is the influenza A virus. The M2 protein of the influenza A virus interacts with MAVS, induces ROS generation, and blocks autophagy to avoid MAVS aggregation, hence dampening the immune response and aiding viral pathogenicity [156]. Following an influenza A virus (IAV) infection as a defense mechanism in normal human nasal epithelial (NHNE) cells, mt-ROS, as well as dual oxidase (Duox)2-ROS, is generated. mt-ROS-mediated signal transducer and activator of transcription (STAT) phosphorylation and IFN-stimulated gene expression is induced by IAV for an anti-viral response. In vitro studies blocking mt-ROS have resulted in an increase in the viral load, thus indicating the crucial role of mt-ROS in the antiviral response [157,158]. On the contrary, the inhibition of mt-ROS in an in vivo mouse model by intranasally injected Mito-TEMPO reduced the viral load, neutrophil infiltration, and lung inflammation [159]. It has also been reported that respiratory syncytial virus (RSV) utilizes enhanced mt-ROS for its proliferation in the absence of complex I. Additionally, the use of mt-ROS inhibitor, mitoquinone mesylate (MitoQ), in the host cell was able to suppress the proliferation of the respiratory syncytial virus (RSV), establishing that RSV utilizes host mt-ROS for its replication and proliferation [160].

The HBx protein of Hepatitis B virus (HBV) localizes in the mitochondria of rat hepatocytes and is capable of modulating transmembrane potential via modulating the transcription factor NF-κB and calcium signaling. Increased calcium signaling increases calcium uptake by cells, causing a calcium plateau. This can be inhibited by blocking store-operated calcium entry (SOCE) and mitochondrial calcium uptake. Blocking SOCE reduces HBV replication, hence establishing a role of calcium in HBV proliferation. It has also been reported that HBx also interacts with the human voltage-dependent anion channel (VDAC). This interaction leads to the depolarization of the membrane potentials [161,162,163].

A regulator of mitophagy, Parkin’s role in the antiviral response is evident from the fact that Parkin knockout was able to reduce the viral load of both RNA [vesicular stomatitis virus (VSV)] and DNA [herpes simplex virus 1 (HSV-1)] viruses. Parkin knockout was also able to enhance mt-ROS-associated NLRP3 inflammasome activity [164].

Hepatitis C virus (HCV) polyproteins hamper the OXPHOS complex I, which leads to enhanced mt-ROS generation, and this is an ultimate effect of enhanced Ca^2+^ uptake in the mitochondria triggered by the interaction with HCV [165].

In severe acute respiratory syndrome coronavirus (SARS-CoV), NLRP3 is activated by the 3a protein of SARS-CoV and mt-ROS, and potassium efflux plays a crucial role in this activation [166]. Rotavirus may alienate the host immune response by initiating ER oxidative stress. The Ca^2+^ ion released from the ER, in turn, increases the mt-ROS generation. This leads to an unfavorable accumulation of ROS for the host cell [167].

A zinc finger antiviral protein (ZAP) PRR has been identified for the RNA of the *Sindbis* viruses, known as tetrachlorodibenzo-p-dioxin (TCDD)-inducible poly (ADP ribose) polymerase (TIPARP). TIPARP induces antiviral responses against the fatal infection of *Sindbis* in mice. The redistribution of TIPARP is dependent on mt-ROS. The localization of TIPARP in cytosol triggers the antiviral response in the host cell [168,169].

Viral proteins can attenuate higher ROS levels by activating the antioxidant system, which is evident in human papillomavirus (HPV) type 16, wherein the constitutive expression of E7 protein led to enhanced catalase expression, hence providing resistance against H_2_O_2_ [170]. Similarly, HCV activates different signaling pathways, including p38 MAPK and JNK via AP-1, and enhances the expression of Mn-SOD (but not Cu/Zn-SOD), initiating an antioxidative mechanism [171].

Apoptosis, a downstream effect of mt-ROS, is avoided by viruses in various ways. One such example is of human cytomegalovirus (CMV), where the viral encoded RNA beta2.7 binds to complex I and hampers the relocalization of subunits in response to an apoptotic stimulus [172]. It has been reported that Vaccinia attenuates the mitochondrial arm of the apoptotic pathway. The attenuation of apoptosis is aided by the F1L protein, which obstructs the reduction in the inner mitochondrial membrane potential and cytochrome c release [173].

Few viruses require the lysis of the cell for the enhancement of infection. One such virus is influenza, which initiates apoptosis by encoding a mitochondrial protein, PB1-F2. This reduces the transmembrane potential and enhances cytochrome c efflux, with the help of ANT-3 and VDAC1 (mitochondrial proteins), which prepares the cell for apoptosis [174].

## 8. Mitochondrial ROS in Fungal Infections

Like bacteria, parasites, and viruses, fungi also elicit an immune response in host cells utilizing mt-ROS. For instance, in *Aspergillus fumigatus* infections, the causative agent for mold pneumonia, conidia, triggers an oxidative burst in the host cell and liberates ROS. Upon *A. fumigatus* infection, different innate immune cells, such as alveolar macrophages, dendritic cells, and neutrophils, generate mt-ROS. Specifically, mitochondrial H_2_O_2_ triggers the anti-fungal response in alveolar macrophages; however, the effects are not so pronounced in dendritic cells and neutrophils. It has also been found that the defects in AM mt-ROS in transgenic mice expressing mitochondria-specific catalase are compensated by NOX and nonoxidative effector mechanisms during *A. fumigatus* infection in mice [175]. In muskmelon, an infection of *Trichothecium roseum* also triggers increased mt-ROS generation, which then produces anti-fungal responses [176]. Elevated mt-ROS is also implicated in acquiring drug resistance, for instance in *Candida glabrata*-azole resistance. A deficiency in the ERMES (ER-mitochondrial encounter structure) component *GEM-1* leads to mt-ROS generation, along with mitochondrial dysfunction and azole efflux pump upregulation, leading to azole resistance [177]. Fungal mitochondria have a completely different take when inducing pathogenicity. In an interesting study, it was reported that treating *Candida albicans* with the cytochrome c oxidase inhibitor sodium nitroprusside (SNP) and the AOX inhibitor SHAM can lead to better recognition by macrophages in vitro and in vivo. However, this pretreatment of SNP+SHAM increased the virulence in vivo. Although an increase in ROS was reported, there was no relation between the increased ROS levels and enhanced hyphal switching in a mouse model [178]. *Cryptococcus neoformans*, an opportunistic yeast, induces pathogenicity via its virulence factor, known as a capsule. The capsule is dynamic in nature and changes its size; usually, it increases its size, and it was reported that this increased size aids *C*. *neoformans* in developing resistance against ROS and avoiding engulfment by phagocytic cells. On the contrary, recently it has been reported that the formation of the capsule requires the accumulation of ROS; however, it is not clear if the accumulation of ROS is required in mitochondria, but enhanced ROS production and an increased membrane potential aid in capsule formation. Additionally, SHAM and AA hamper the formation of the capsule [179,180]. In *C. neoformans*, heat shock factor 3(CnHsf3) regulates mt-ROS homeostasis by increasing ETC gene expression and reducing the expression of TCA cycle genes [181]; hence, this could be a target for therapy. Previously, it was reported that, under hypoxic conditions in the host, *Candida albicans* could mask β-glucan via an elevation in mt-ROS [182]. Additionally, the AA-induced inhibition of the bc1 complex of *Candida albicans* increased ROS generation (though it is not specified which ROS); consequentially, higher levels of unsaturated fatty acids are produced, altering the cell membrane structure and modifying the cell wall structure. The β-glucan becomes more accessible to the macrophages, improving their phagocytic efficacy [183]. *Aspergillus fumigatus*, which causes invasive pulmonary aspergillosis, demonstrates a fascinating interplay among mitochondrial ETC complexes and oxidative stress. Under the host’s hypoxic condition, the respiratory chain of *A. fumigatus* remains active, aiding in the virulence. Mutants were created for two components, AOX (▲aoxA) and cytochrome c (▲cycA), in *A. fumigatus.* ▲aoxA was more susceptible to extracellular ROS than ▲cycA, notwithstanding the fact that ▲aoxA was more virulent than ▲cycA. It was established that AOX aids in resisting the host’s ROS and its cytochrome c aids in initiating the virulence, although the mechanism is not completely understood; a possible argument could be that regulation of ROS governed by cytochrome c in *A. fumigatus* could help in recognizing the hypoxic conditions in the host and enhance the virulence [184]. However, in *C. neoformans*, AOX gene; AOX-1 mutants were less virulent, as well as more sensitive to exogenous stress [185].

With the reviewed data to this point, we are able to affirm that mt-ROS and mitochondrial processes play a crucial role in animal immunity, but Yang et al. have shown that mt-ROS is significant in plant immunity as well. They investigated the role of the pentatricopeptide repeat (PPR) protein of *Arabidopsis thaliana*, called RESISTANCE TO PHYTOPHTHORA PARASITICA 7 (RTP7), which is crucial for the posttranscriptional RNA splicing of mitochondrial genes. It was found that RTP7’s role in maintaining immunity was conserved across distantly related plant species against a plethora of pathogens. RTP7 usually splices nad7, a complex I component. In RTP7 mutants, this RNA processing of complex I was altered, leading to the enhanced production of mt-ROS. In these mutants, mitochondria accumulate around the haustorial neck with elevated levels of mt-ROS potentially attenuating pathogen infection. For instance, an RTP7 mutant of *A. thaliana* showed high resistance against the oomycete *Phytophthora parasitica* by increasing mt-ROS levels. This is evident, as the utilization of the mt-ROS-specific scavenger MitoTEMOPL and the antioxidant NAC increased the susceptibility of *A. thaliana* significantly towards *Phytophthora parasitica.* In RTP7 mutants, AOX levels were also upregulated. Other pathogens against which it showed resistance are the bacterial pathogen *Pseudomonas syringae pv. tomato (Pst)* DC3000 and fungal pathogens like *Botrytis cinerea* and *Rhizoctonia solani* [186].

## 9. Future Perspectives: Targeting Mitochondrial ROS as a Therapeutic Approach

Mitochondrial reactive oxygen species (mt-ROS) generated by the mitochondria are widely applicable in a number of physiological processes, ranging from transcription to translation, apoptosis, and immunity. Of various pathogen groups, we have discussed how bacteria, viruses, parasites, and fungi interact with the host and how mt-ROS govern the immunological responses. mt-ROS, in general, elicit immunological responses through the activation of NLRP3 inflammasome. Bacteria adopt various mechanisms to disrupt the host mt-ROS in order to evade the host response; however, it has also been shown that increased mt-ROS levels also aid bacteria in prolonging their existence, as evident from *Acetobacter baumannii* and *Pseudomonas aeruginosa*. Dreaded parasites like *Leishmania*, *Trypanosoma*, *Plasmodium*, or *Toxoplasma* exploit various pathways; for instance, *T. cruzi* induces the fibrotic gene expression pathway through enhanced mt-ROS production. *T. gondii*, on the other hand, hampers the OXPHOS pathway. *L. donovani* have been reported to enhance the antioxidant mechanism to increase their survival inside the host cell. Viruses, as intracellular parasites, modulate host machinery and regulate mt-ROS levels to evade the host immune response. Fungi have the capability to mask their cell wall components to avoid the host immune response and attain drug resistance. Overall, this review gives a concise overview of the mechanisms adopted by the host and pathogens, which could give a newer direction to the therapeutic approach.

By now, we have a fair understanding of how mt-ROS vary under different conditions and how various pathogens either modulate their own pathways or exploit the host environment to regulate mt-ROS, in order to establish a successful infection. This adaptation or modulation of the surroundings could lead to a higher risk of toxicity and resistance in most of the infectious diseases caused by bacteria, viruses, fungi, or protozoans [187,188,189,190]. To alleviate this issue, it is imperative to find newer drug targets or repurpose existing drugs that might switch the modulation of mt-ROS towards reinforcing the host immune response. Since we are focusing exclusively on mt-ROS, we will give a brief overview of the drugs which target or have the capability to modulate mt-ROS.

The most convenient solution would be to use naturally occurring phytochemicals. As described earlier, bacteria can induce mt-ROS and initiate NLRP3 inflammasome formation. To alleviate this issue, studies have been carried out with the naturally occurring phytochemical quercetin. The utilization of quercetin was able to reduce mt-ROS generation while maintaining the mitochondrial function and to attenuate NLRP3 activation while inducing autophagy upon infection by *E. coli* O157:H7 [104]. However, the same quercetin, when used in combination with azithromycin, was able to increase ROS and induce mitochondrial damage, which helps with the inhibition of *Toxoplasma gondii* tachyzoite [191]. Utilizing another widely used phytochemical, curcumin, resulted in anti-leishmanial properties by inducing calcium efflux and the depolarization of membrane potential, and increasing ROS [192]. Curcumin was also found to be effective as an antifungal agent against *vulvovaginal candidiasis*. Curcumin was able to reduce the vaginal fungal burden in rats due to *C. albicans* [193]. Barberine is another phytochemical whose ability to induce oxidative stress and mitochondrial dysfunction in cancer is well documented. Barberine chloride resulted in a dose-dependent reduction in *Leishmania parasites.* It was also implicated in the enhancement of ROS and mt-ROS, along with the depolarization of the mitochondrial membrane potential, and the inhibition of mitochondrial complexes I-III and II-III in UR6 promastigote. It also reduced ATP levels, indicating mitochondrial dysfunction in *Leishmania* [194].

Another method could be the utilization of nanoparticles. As explained previously, nanoparticles are also a stimuli which generate ROS. Apart from that, another advantage of using nanoparticles would be that they are equal in size or could be made to be a similar size which could be easily taken up by the organs and tissues; immune responses are triggered because these are recognized as foreign particles by the immune system and may form the basis of future vaccine designs [195]. For instance, curcumin-loaded nanostructures have been reported to exhibit leishmanicidal effects [196]. Apart from these, as previously discussed, the inhibitors of respiratory chain complexes or the sites for the release of electrons could be potential therapeutic targets. For instance, a well-known drug used against *Leishmania* infection is camptothecin (CPT), which targets the nuclear DNA topoisomerase I enzyme. Interestingly, it has been observed that, in cancer cells, mitochondria dual-targeting polyprodrug nanoreactors (DT-PNs), covalently attached with a high content of repeating camptothecin (CPT), induce mt-ROS upregulation, resulting in cancer cell apoptosis [169]. The same strategy could be employed to treat many intracellular pathogens, like *Leishmania*, which downregulate cellular ROS production to their advantage.

## 10. Conclusions

With an unpaired electron and a mandatory oxygen atom, ROS are the most crucial entity for the maintenance of the redox balance of the cell. These ROS are generated by different organelles and compartments of the cell, such as the cytoplasm, ER, peroxisome, and mitochondria. However, an unregulated amount of ROS may lead to cellular damage. Mitochondria are of great importance as they not only generate ROS but also have an antioxidant system of their own. The antioxidants are not just capable of scavenging mt-ROS but also eliminate external ROS, indicating that mitochondria are the potential sink for the majority of ROS generated [6,44]. In conclusion, the intricate balance between reactive oxygen species (ROS) as both detrimental agents, causing oxidative stress, and essential modulators of physiological processes highlights the complex interplay within cellular systems. The dynamic role of mitochondrial ROS (mt-ROS) in regulating key pathways of host immune responses during infections by various pathogens underscores their multifaceted impact on host–pathogen interactions. Understanding how pathogens manipulate mt-ROS to evade host immunity opens new avenues for exploring potential therapeutic strategies targeting these mechanisms in infectious diseases. This review sheds light on the crucial role of mt-ROS in immune responses and pathogen evasion strategies, paving the way for further research and therapeutic developments in this field.

## Figures and Tables

**Figure 1 biomolecules-14-00670-f001:**
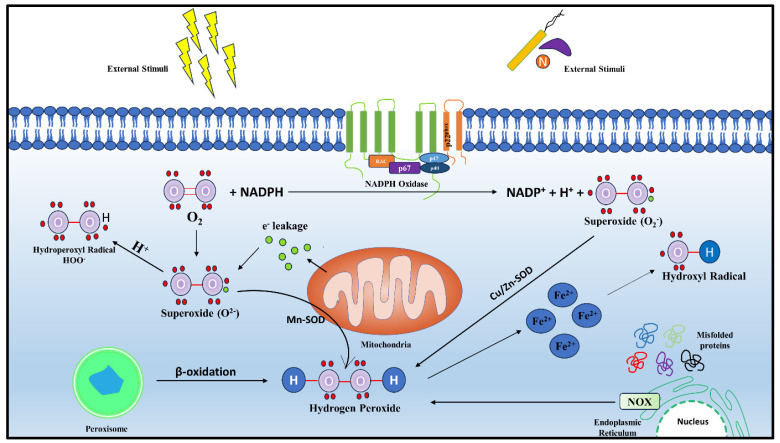
Reactive oxygen species (ROS) generation at different subcellular locations. Reactive oxygen species (ROS) are generated by various organelles across the cell upon being stimulated by external stimuli like pathogens, radiation, or nanoparticles. NADPH oxidase (NOX), with the help of its cytosolic subunits p67, 47, and 40, and membrane proteins like p22^phox^, converts NADPH to NADP^+^ and generates superoxide. Β-oxidation taking place in the peroxisome yields hydrogen peroxide, which is also formed during ER stress due to misfolded proteins, where ER-specialized NOX aids in ROS generation. When hydrogen peroxide reacts with Fe^2+^, it undergoes Fenton’s reaction and produces hydroxyl radicals, while the interaction of superoxide with protons yields hydroperoxyl radicals. For mitochondria, the major contributor of ROS is the electron transport chain.

**Figure 2 biomolecules-14-00670-f002:**
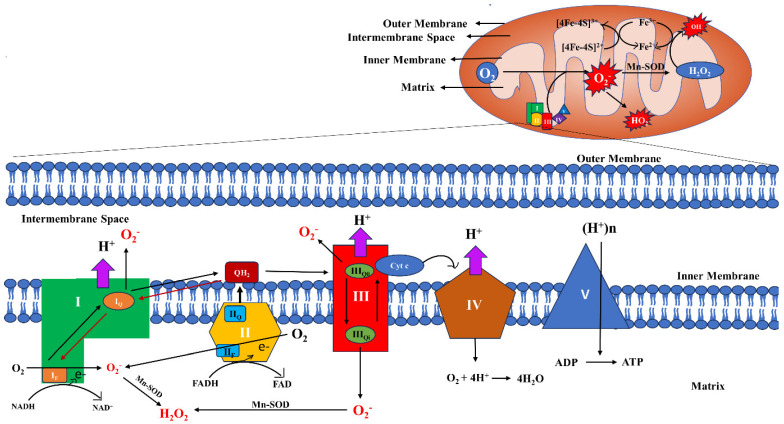
Electron leakage leads to superoxide formation. The superoxide formed in mitochondria is converted to hydrogen peroxide by SOD enzyme in the mitochondria, which is later converted to water by antioxidants like catalase. Reverse electron transfer shown by red arrow also aids in ROS generation. Iron–sulfur clusters also aid in ROS generation.

**Figure 3 biomolecules-14-00670-f003:**
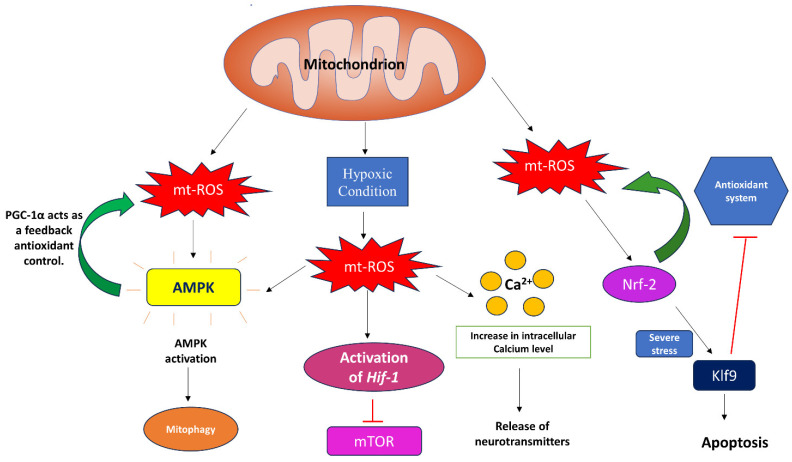
mt-ROS signaling. mt-ROS is crucial in many signaling pathways. Here, in the diagram, we have shown how, under hypoxic conditions and normal physiological conditions, mt-ROS aids in regulation of transcription factors, release of neurotransmitters, apoptosis, mitophagy, and antioxidant system.

**Figure 4 biomolecules-14-00670-f004:**
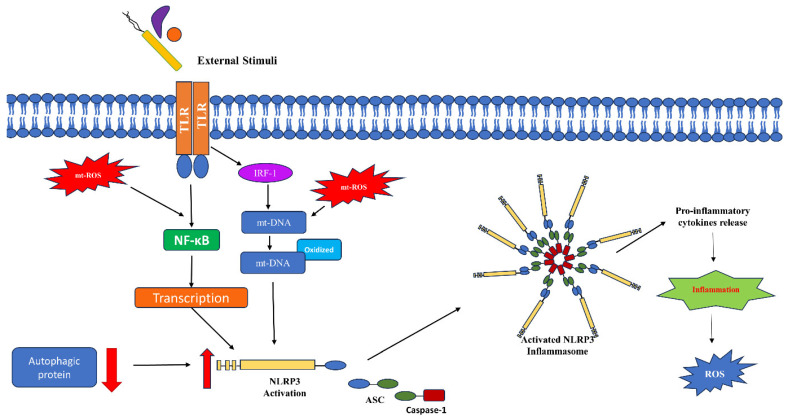
mt-ROS in NLRP3 inflammasome formation. mt-ROS aids NLRP3 formation in the following ways—(1) by aiding the transcription of NF-κB, which helps to overcome the naturally low levels of NLRP3 sensors. (2) By oxidizing the mt-DNA synthesized via TLR pathway, which directly binds to NLRP3 and activates it. (3) Through the deficiency of autophagic proteins, such as beclin-1, enhancing the mt-ROS generation and consequently NLRP3 activation. Upon activation, NLRP3 produces proinflammatory cytokines, which ultimately lead to ROS accumulation.

**Table 1 biomolecules-14-00670-t001:** Effects on and of mt-ROS on host and pathogens. (+ = beneficial, − = detrimental, +/− = previously beneficial and then detrimental, −/+ = detrimental at the beginning and then beneficial. ↑ shows the increase in mt-ROS and ↓ shows the decrease in mt-ROS.

	Pathogenic Microbes	Model	Effect on mt-ROS	Effect on		Action	Reference
	Bacteria			Host	Pathogen		
(1)	*A. baumannii*	Beas 2B cells; RAW264.7cells	↓ ↑	+ +/−	− −/+	L-serine decreases the virulence of *A. baumannii*. Inhibits NLRP3 inflammasome activation by decreasing mt-ROS and total ROS. Outer membrane protein 34 of *A. baumannii* induces mt-ROS production and activates NLRP3. Prolonged activation may lead to severe infection.	[102,103]
(2)	*E. coli* O157:H7	Caco-2 cells	↑	−	+	Bacteria induces high degree of NLRP3 inflammation.	[104]
(3)	*Enterococcus faecalis*	*C. elegans*	↑	+		isp-1 mutation in *C. elegans*, which reduces mitochondrial respiration, increases mt-ROS via AMPK and Hif-1 pathways and aids in attaining resistance against *E. faecalis*.	[105]
(4)	*P. aeruginosa*	Neutrophils	↑	−	+	Bacterial toxin pyocyanin; releases cytochrome c, induces mt-ROS, activates sphingomyelinase, and aids in neutrophil elimination.	[106]
(5)	*P. gingivalis*	Gingival epithelial cells (GECs)	↓	−	+	Attenuates eATP/NOX-2-ROS pathway, which is one of the antibacterial pathways.	[107]
Protozoan parasites:							
(6)	*T. cruzi*	C57B6 WT mice. HeLa cells, C57B6 MnSODtg	↑ ↓	− +	+ −	Attenuates NFE2L2/ARE pathway induces fibrotic gene expression and leads to cardiomyopathy. Activates NFE2L3/ARE pathway reduces fibrotic gene expression, inhibits cardiac failure.	[108]
(7)	*T. cruzi*	Epimastigotes	↑	NA	+	Mitochondrial membrane potential is modulated via heme, which induces mt-ROS generation and aids in proliferation and survival of parasites.	[109]
(8)	*Toxoplasma gondii*	Human foreskin fibroblast (HFF) cells	↑	−	+	Hampers host mitochondrial function for its survival. It includes deregulation of OXPHOS, production of mt-ROS, and reduced expression of complex IV, I, and II, respectively.	[110]
(9)	*Plasmodium berghei*	*Anopheles gambiae*	↓	−	+	AgMC1 in *Anopheles gambiae* mosquitoes reduces membrane potential and increases proton leak and uncoupling, which ultimately reduces mt-ROS in midgut and increases the plasmodium susceptibility.	[111]
(10)	*Plasmodium vivax*	Monocytes	↑	+		There is a metabolic shift in the monocytes. Glycolysis is preferred during the infection for ATP generation. This dysregulates the mitochondrial functions and increases mt-ROS. While the mt-ROS performs the protective role, it also induces Mn-SOD, which aids in avoiding tissue damage.	[112]
(11)	*Leishmania donovani*	Murine macrophages	↓	−	+	Macrophage cholesterol and mt-ROS levels are manipulated by *L. donovani* via the SREBP2 pathway. This leads to UCP2 upregulation and the downregulation of mt-ROS, which aids parasite survival. This also inactivates MAPK to attenuate Th1 response.	[113,114]
Viruses							
(12)	Respiratory syncytial virus (RSV)	A549 cells, vero cells, BCi-NS1 cells, pBECs	↑	−	+	It induces microtubule/dynein-dependent mitochondrial perinuclear clustering, and translocation towards the microtubule-organizing centre. This hampers mitochondrial functions and membrane potential and increases mt-ROS, which aids in the infection.	[115]
(13)	Human immunodeficiency virus (HIV)	Astrocytes	Productive infection: ↓ Abortive infection:↑	+	+	With reduced mt-ROS, cell death resistance occurs. Mitochondrial damage and inflammasome activation.	[116,117]
(14)	Enterovirus 71 (EV71)	SF268 glioblastoma cells	↑	-	+	Decrease in ATP production. Increased mt-ROS. mtROS activates PKR and PERK, which in turn activates eIF2a phosphorylation, crucial for viral replication as inhibiting this reduces viral titer.	[118]
Fungi:							
(15)	*Aspergillus fumigatus*	Murine bone marrow-derived macrophages (BMDMs)	↑	+	-	mt-ROS is produced via RET and complex II. Aids in cytokine production. Has an antifungal effect.	[119]

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
