# Peer review of "Mitochondrial Reactive Oxygen Species in Infection and Immunity"

_biomolecules, 2024, doi:10.3390/biom14060670_

Round 1

Reviewer 1 Report

Comments and Suggestions for Authors

The manuscript will improve a lot after some revisions as to its flow and organization. In some cases, the idea and story becomes difficult to follow because of lack of ample introduction to the subtopic, sudden jumps to a new topic. Please consider the following points:

1. The importance of discussing mtROS during pathogenic infection is not well described. 

2. I do not see the relevance of the part "In plants lower concentration of ROS favours root hair formation and a higher concentration is required for the polar growth of root hair (6)."

3. The part " Generation and regulation of mitochondrial ROS" is too long and could be divided into smaller subtopics. 

4. I do not think the A. thaliana mtROS is necessary per se to include in the discussion as the focus of the work should mostly highlight those in infection and immunity. 

5. The part "Signalling through mitochondrial reactive oxygen species" is jumpy. It should be divided into several sections.

6. Authors should add an introductory part for the part "Mitochondrial ROS in immune signalling".

7. I also think the authors should consider adding another figure summarizing the role of mtROS in signaling.

8. Table 1 title is obviously wrong. Also consider using landscape format to improve readability of the table.

Comments on the Quality of English Language

Quality of English is good.

Reviewer 2 Report

Comments and Suggestions for Authors

Reply to Editor: Mukherjee A et al. Mitochondrial ROS in infection and immunity.

The work of Mukherjee et al. certainly represents an interesting challenge and it is to be welcomed that the colleagues have taken on this task. The detailed description of signaling chains, especially in the function of the immune system and in particular with regard to bacterial and viral infections, as well as protozoan infections, should be emphasized. The reviewer understands the structure of the work as a sequence of an initial basic explanation of processes (OXPHOS etc.) of ROS formation and then a special section on pathophysiological processes. In this sense, however, fundamental problems arise for the reader:

The  article is about "MITOCHONDRIAL (!) ROS IN INFECTION AND IMMUNITY". However, after the introductory basic part of the paper, the authors describe various activation pathways that lead TO the mitochondria, but leave the direct mechanisms in the mitochondria in the background.

See e.g: Okoye CN et al, Function Complex I in Ischemia, Significance of Complex II in REF, if applicable, etc.

A statement such as on page 4 "Subsequently,.....a.s.o." to describe the ETC function is clearly not enough. Moreover, on page 2 it says: "In a resting cell......"! This wording raises big questions! When do we have a "resting cell"? In the heart, during permanent contraction and relaxation, i.e. energy consumption, in the liver, during detoxification, glucose metabolism etc., in the kidneys, during energy-consuming processes of ion exchange? Sorry, that's not clear.

Therein we have the problems of the first sections of the article. Individual mechanisms and processes in the ETC and interactions with the mitochondrion are listed, but there is no reference to compartmentalization in the cell and no clear distinction between physiological processes and a "cell under stress" (e.g. ischemia or hypoxia). These are two different viewpoints!

It is stated (page 4) that approx. 0.2 to 2% of the electrons from the ETC contribute to the production of ROS with reference to the work of Martin Brand. But even he avoids a subdivision between different activity states in the ETC.  If one starts a discussion of this way, then one must also cite pros and cons for mitochondria as the main source of ROS (Zhang et al. J Exp Biol 2021, 224, 221606) AND explain the substrate dependence of ROS production. Unfortunately, these points are missing. The reviewer refers to the article by Napolitano G et al. Antioxidants 2021, 10, 1824.

Last but not least, the reviewer refers to the redox regulation of respiratory chain enzymes (Povea-Cabello et al. BST 2024, 52, 873).

It is therefore advisable to fundamentally revise the beginning sections. The subsequent statements on infection and the functional description of the immune system would then follow in an absolutely comprehensible manner.

Ref.:

Zhang Y, Wong HS. Are mitochondria the main contributor of reactive oxygen species in cells? J Exp Biol. 2021 Mar 11;224(Pt 5):jeb221606. doi: 10.1242/jeb.221606. PMID: 33707189.

Okoye CN, Koren SA, Wojtovich AP. Mitochondrial complex I ROS production and redox signaling in hypoxia. Redox Biol. 2023 Nov;67:102926. doi: 10.1016/j.redox.2023.102926. Epub 2023 Oct 16. PMID: 37871533; PMCID: PMC10598411.

Povea-Cabello S, Brischigliaro M, Fernández-Vizarra E. Emerging mechanisms in the redox regulation of mitochondrial cytochrome c oxidase assembly and function. Biochem Soc Trans. 2024 Apr 24;52(2):873-885. doi: 10.1042/BST20231183. PMID: 38526156.

Napolitano G, Fasciolo G, Venditti P. Mitochondrial Management of Reactive Oxygen Species. Antioxidants (Basel). 2021 Nov 17;10(11):1824. doi: 10.3390/antiox10111824. PMID: 34829696; PMCID: PMC8614740.

Comments on the Quality of English Language

fine

Reviewer 3 Report

Comments and Suggestions for Authors

Biomolecules-3035875

The manuscript entitled “Mitochondrial reactive oxygen species in infection and immunity” submitted by Arunima Mukherjee, Krishna Kanta Ghosh, Sabyasachi Chakrabortty, Balázs Gulyás, Parasuraman Padmanabhan and Writoban Basu Ball to be edited by Biomolecules is very ambitious and quite heavy sometime.

The submitted manuscript is very long and can probably be condensed without losing any information. However, with regard to mitochondrial ROS, the authors devote too little space to the role of ROS in inducing apoptosis and virtually ignore autophagy.

The text lacks a chapter on mitochondrial ROS inhibitors, line MitoTEMPO and more particularly the work of Vladimir skulachev's laboratory on SKQ1 and associated molecules.

But over this details that could be corrected, I don’t see any reason to refuse this manuscript from being published.

Major remaks:

• Talking about mitochondrial ROS, apparently the authors did forget that the main products are the superoxide anion (O2.-) and the supsequent hydroperoxide (H2O2).

Minor remarks:

 • H2O2 should be written like H2O2

Round 2

Reviewer 2 Report

Comments and Suggestions for Authors

I thank You for your reply. Although You have not mentioned the ATP- dependent regulaton of Cytochrome Oxidase relevant for ROS in the introducing paragraphs the ms has won more quality. Congratulations and go ahead.